Weak population structure in the ant Formica fusca

Johansson Helena 1 helena.z.johansson@helsinki.fi
Seppä Perttu 1 2
Helanterä Heikki 1 2
Trontti Kalevi 1
http://orcid.org/0000-0002-6176-8940 Sundström Liselotte 1 2
1 Centre of Excellence in Biological Interactions, Organismal and Evolutionary Biology Research Programme, University of Helsinki , Helsinki , Finland
2 Tvärminne Research Station, University of Helsinki , Hangö , Finland
Hughes Jane
Electronic publication date: 2018 Jun 19
Publication date: 2018
Volume: 6
Electronic Location ID: e5024
Received 2018 Jan 8; Accepted 2018 May 26
Copyright: © 2018 Johansson et al.
Copyright year: 2018
Copyright holder: Johansson et al.
License: This is an open access article distributed under the terms of the Creative Commons Attribution License, which permits unrestricted use, distribution, reproduction and adaptation in any medium and for any purpose provided that it is properly attributed. For attribution, the original author(s), title, publication source (PeerJ) and either DOI or URL of the article must be cited.
License URL: https://creativecommons.org/licenses/by/4.0/

Keywords: Sex-biased dispersal, Formica fusca, Panmixia, Social insect, Microsatellite

Funding: Finnish Cultural Foundation (HH) Academy of Finland project-numbers (LS) Centre of Excellence in Biological Interactions number 252411, and 284666 LS grant numbers 121216 and 251337 HH grant numbers 121078, 135970 This study was funded by the Finnish Cultural Foundation (HH) and the Academy of Finland project-numbers: (LS) Centre of Excellence in Biological Interactions number 252411, and 284666, and LS grant numbers: 121216 and 251337, and HH grant numbers 121078, 135970. The funders had no role in study design, data collection and analysis, decision to publish, or preparation of the manuscript.

==============================
Dispersal is a fundamental trait of a species’ biology. High dispersal results in weakly structured or even panmictic populations over large areas, whereas weak dispersal enables population differentiation and strong spatial structuring. We report on the genetic population structure in the polygyne ant Formica fusca and the relative contribution of the dispersing males and females to this. We sampled 12 localities across a ∼35 km2 study area in Finland and generated mitochondrial DNA (mtDNA) haplotype data and microsatellite data. First, we assessed queen dispersal by estimating population differentiation from mtDNA haplotype data. Second, we analysed nuclear DNA microsatellite data to determine overall population genetic substructure in the study area with principal components analysis, Bayesian clustering, hierarchical F statistics and testing for evidence of isolation-by-distance. Third, we directly compared genetic differentiation estimates from maternally inherited mtDNA and bi-parentally inherited DNA microsatellites to test for sex-bias in dispersal. Our results showed no significant spatial structure or isolation by distance in neither mtDNA nor DNA microsatellite data, suggesting high dispersal of both sexes across the study area. However, mitochondrial differentiation was weaker (Fst-mt = 0.0047) than nuclear differentiation (Fst-nuc = 0.027), which translates into a sixfold larger female migration rate compared to that of males. We conclude that the weak population substructure reflects high dispersal in both sexes, and it is consistent with F. fusca as a pioneer species exploiting unstable habitats in successional boreal forests.

Introduction

Dispersal serves to move genes between natural populations, maintaining gene flow and connectivity, thereby reducing the risks of local population extinction and genetic drift (Clobert et al., 2001). High gene flow may lead to weak population substructure or even panmixia over large areas (Peel et al., 2013), whereas reduced gene flow often leads to significant substructure and divergence between populations, through genetic drift or through strong selection acting on the diverging populations (Slatkin, 1987). Genetic population substructures may also arise due to differential dispersal in e.g. sexes or morphs (Seppä et al., 2006) and reveal social relationships, especially in highly social and eusocial species (Ross, 2001). The scale at which population genetic patterns arise can yield information about the processes that leads to divergence, as well as a species potential migratory or dispersal ability (Kovach et al., 2013).

Eusocial insects (ants, termites, some wasps, and bees) form societies in which only specialized sexual castes (queens and males) disperse and reproduce, whereas workers forgo their own reproduction to help raise the offspring of the resident queen(s). In ants, queens and males are commonly winged (Peeters & Ito, 2001) and flight is the primary medium for mating and gene flow (Helms, 2018). However, ants are slow fliers (1.3–5 m/s) and there is great variation between species in distance covered (30–32,200 m), duration (25–140 min) and altitude of flight (1–300 m) (Helms, 2018). These three factors are influenced by physiology of the flying individuals, such as size, wing size, fat storage capacity, and behaviour. Ant species with small queens that have relatively large wings and fat content (>40% of their body weight) tend to fly at higher altitudes and cover the largest distances (Helms, 2018, and references therein). Furthermore, dispersal capability and distance are associated with the nest-founding strategy of queens. Queens able to found nests independently, through temporary parasitism, or by joining unrelated queens in foundress associations, can usually disperse relatively far (Keller & Passera, 1989). Conversely, queens that rely on the aid of their own workers to found their nests (dependent nest founding) or rely on re-adoption into established colonies are restricted in their dispersal (Keller & Passera, 1989; Cronin et al., 2013). Dependent founding is commonly found in species with several related queens in a single colony (polygyne species), and is thought to be an adaptation to reduce dispersal risks (Keller & Passera, 1989; Cronin et al., 2013).

Queen founding strategy has consequences for hierarchical genetic substructure, i.e. partitioning of genetic variation from nests to populations or species. With independently founding queens, the resulting population substructure is expected to show strong structure at the nest level, yet sub-structuring at higher hierarchical levels such as habitat patch or population may be absent if queen dispersal is strong. With dependently founding queens short-distance dispersal is expected to lead to genetic viscosity (isolation-by-distance (IBD) over small spatial scales) and discontinuous population substructure (Ross & Shoemaker, 1997; Ross, 2001). Dependently founding species may also form polydomous populations, i.e. comprise networks of related and interconnected nests that sometimes occupy entire habitat patches (Pamilo et al., 2005). Consequently, strong substructure at the level of the patch is expected, and substructure at the level of a population is often observed (Pamilo et al., 2005; Holzer, Keller & Chapuisat, 2009). The strength of these patterns is modified by the rate of male dispersal (see e.g. Sanllorente, Ruano & Tinaut, 2015).

The population level consequences of the queens’ strategy for nest founding can be elucidated from patterns of population differentiation based on maternally inherited mitochondrial DNA (mtDNA). Furthermore, by contrasting mtDNA differentiation with that estimated from nuclear markers (e.g. DNA microsatellites) relative levels of female and male dispersal can be estimated. Such research on ants has revealed that dispersal is usually but not overtly male biased, with greatest bias in species with wingless queens, regardless of gyny level (Table S1; Sundström, Seppä & Pamilo, 2005). To our knowledge, female-biased dispersal has not been inferred with genetic data in any study on ants.

While different population genetic consequences between independently and dependently founding species has support from empirical data (Ross & Shoemaker, 1997; Ross, 2001), species with both modes of founding are more challenging. Unless queens lack wings entirely, winged (alate) queens remain potentially important for dispersal even if dependent founding is common, i.e. a few dispersing queens may be sufficient to allow efficient gene flow to homogenize populations (Peeters & Ito, 2001). Both modes of founding may co-exist in the same population, or one may become dominant in different populations within the same species, such as in Formica exsecta (Sundström, Seppä & Pamilo, 2005). Population substructures in such species are also less predictable; they may change with the number of reproductive queens in the colony (i.e. one (monogyne) or several (polygyne) queens per nest)), as has been shown in F. truncorum (Sundström, 1995), or, like in the case of F. polyctena, spatial scale (Gyllenstrand, Seppä & Pamilo, 2004). Studying such species may hence offer an insight into ant dispersal evolution, and the social and ecological conditions under which dispersal risk adaptations are likely to evolve.

European members of the genus Formica have long been an important model for population genetics, dispersal and ecology in ants. The genus is socially diverse and includes dependently founding species with obligate polygyny, polydomy and strong population genetic structuring, such as F. aquilonia (Pamilo et al., 2005) and F. paralugubris (Holzer, Keller & Chapuisat, 2009), but also independently founding species (with or without queen re-adoption), such as F. rufa (Gyllenstrand, Seppä & Pamilo, 2004), socially polymorphic species with both types, such as F. exsecta (Pamilo & Rosengren, 1984) and F. truncorum (Sundström, 1993), and species with continuous variation in queen numbers, such as F. fusca (Bargum, Helanterä & Sundström, 2007). The genus is also ecologically diverse, inhabiting a wide variety of habitats from sand dunes (F. cinerea) to mature conifer forest (F. rufa group) (Czechowski, Radchenko & Czechowska, 2002). Habitat stability has been proposed to promote evolution of dependent founding and polydomy in Formica species (Rosengren & Pamilo, 1983), yet no study has directly addressed the population genetic consequences of inhabiting temporally unstable habitat in this group of ants. To date, male-biased dispersal has been confirmed in F. paralugubris (Holzer, Keller & Chapuisat, 2009), F. exsecta (Seppä et al., 2004) and F. lugubris (Gyllenstrand & Seppä, 2003) and thought to be a consequence of polygyny.

Here, we investigate patterns of population differentiation in the widespread generalist ant F. fusca (Czechowski, Radchenko & Czechowska, 2002) which is one of the first species to arrive in clear-cut areas in boreal forests (Punttila et al., 1991). Both sexes are winged and mature to flight in July, but patterns of nuptial flight and mating behaviour are unknown. It founds new colonies independently (Pamilo, 1983), suggesting that dispersal distances can be substantial. The majority of nests (55–68%) are polygyne (Hannonen, Helanterä & Sundström, 2004; Bargum & Sundström, 2007) and 26% of queens (regardless of gyny level) are multiply mated, resulting in worker relatedness levels around 0.65 in monogyne nests (Hannonen, Helanterä & Sundström, 2004; Bargum & Sundström, 2007). Nest-mate queen relatedness values in polygyne nests range from 0 to 0.8 suggest that in some nests daughter queens are re-adopted, and in other nests the queens are unrelated (Hannonen & Sundström, 2003; Hannonen, Helanterä & Sundström, 2004). Previous small-scale studies on island populations (≤5 km2) using DNA microsatellites and allozymes suggest weak population substructure, no dispersal limitation, monodomy (nests do not form interrelated networks), and no genetic differentiation between social forms (Pamilo, 1983; Bargum, Helanterä & Sundström, 2007).

We investigated genetic population substructure in a continuous area with principal component and Bayesian assignment methods, and complemented these with tests of hierarchical population substructure, and an IBD analysis. We were asking whether population substructure remains weak also over larger spatial scales in a homogenous landscape, or if we can observe a change towards greater genetic structure with increasing distance due restricted dispersal. To test for sex-biased dispersal, and to assess the sex-specific contribution to population genetic structure, we exploited the different properties of microsatellites (bi-parental, larger effective population size) and mtDNA (maternal inheritance, smaller effective population size) by estimating and comparing population differentiation in these two markers.

Materials and Methods

Sampling and molecular methods

A total of 12 localities were sampled in April–June 2005 in an area spanning ca. 10 × 35 km in Southern Finland (Fig. 1; Table 1). Based on vegetation characteristics the localities varied in age, however no formal assessment of age was made. The locality altitudes, the monthly mean and maximum temperatures and precipitation values, as well as the bioclimate values (http://www.worldclim.org/bioclim, Table S3) suggest homogenous bioclimatic conditions across this area. There are no major geographical barriers to dispersal in this region (Fig. 1), nor are there post-glacial contact zones potentially contributing to the estimated population differentiation and dispersal estimates (Schmitt, 2007). At each sampling locality, five workers and all queens found were collected from between five and 15 nests. For all localities, the exact location for the first nest that was sampled was recorded with a Garmin GPS 12XL, and for the subsequently sampled nests coordinates were calculated from the metric distance and angle.

Figure 1 Map of sampling localities in southern Finland.

Haplotypes found and neighbour-joining tree of the haplotypes shown. Each filled circle represents a nest; a group of filled circle represents a sampling locality. The colour of the filled circle matches the colour in the NJ tree and shows which haplotype was found in the nest.

Table 1 Locality names, locality short names (LOC), latitude, longitude, numbers of individuals typed at a minimum of four microsatellite loci (NIND), numbers of colonies (NCOL) Genetic diversity HS, allelic richness AR, FIS, and number of haplotypes (NHAP).

Name	LOC	Latitude	Longitude	NIND	NCOL	HS	AR	FIS	NHAP	
Bäckvägen	BAC	59.8438629	23.1559805	73	15	0.468	2.942	0.061	3	
Byvägen	BYV	59.8447971	23.1896538	37	8	0.481	3.033	−0.071	2	
Forcit	FOR	59.8609232	23.0380195	24	5	0.388	2.473	0.187	3	
Grabbskog	GRA	60.0473469	23.4136252	25	6	0.408	2.604	−0.082	2	
Gråkärr	GRK	60.0268888	23.3718594	26	6	0.455	2.833	−0.13	2	
HankoP	HAN	59.8411238	22.9778762	26	6	0.538	3.367	−0.062	2	
Långvik	LAN	59.9320293	23.3350602	47	10	0.505	3.146	0.056	3	
Mattakärr	MAT	59.923116	23.2513624	60	13	0.522	3.206	−0.038	5	
Mossåker	MOS	59.8460816	23.1682066	15	3	0.557	3.311	0.196	2	
Öby	OBY	59.9147305	23.2125462	39	6	0.467	2.798	−0.069	3	
Pojo	POJ	60.0328783	23.4600863	42	9	0.569	3.369	0.079	4	
Sandö	SAN	59.8608072	23.0791277	29	6	0.386	2.703	−0.172	2	
Totals/averages	–	–	–	443	93	0.479	2.982	0.01	2.75	
Note:

Totals are given for number of individuals and colonies, averages are given for the genetic diversity measures and number of haplotypes.

All workers were genotyped at nine DNA microsatellite loci: FL12, FL20 (Chapuisat, 1996), FE13, FE17, FE19, FE21 (Gyllenstrand, Gertsch & Pamilo, 2002), FY4, FY7, and FY13 (Hasegawa & Imai, 2004), following protocols within. Genotyping was performed on a MegaBACE 1000 with the internal size standard ET400-R (GE Healthcare, Little Chalfont, UK) and scored using Fragment Profiler v1.2 (GE Healthcare, Little Chalfont, UK).

For one queen (preferentially) or one worker from each nest, a ca. 600 bp segment of the mitochondrial COI gene was amplified with primers: coi_bar_F: 5′-ACTAGGATCTCCAGACATAGC-3′ and coi_bar_R: 5′-GCTCGTGTATCAACATCTAA-3′, following protocols in Seppä et al. (2011). Sequencing of the PCR-products was carried out by the Institute of Biotechnology DNA Sequencing and Genomics Laboratory (University of Helsinki, Finland).

Analyses on the complete data set

For the DNA microsatellite data we tested for population deviations from Hardy–Weinberg equilibrium (HWE) and linkage disequilibrium (LD) using Arlequin (Excoffier & Lischer, 2010). Allele frequencies averaged over all populations, local and global inbreeding coefficients (FIS), gene diversity (HS) and allelic richness (AR) were calculated for each locality using Fstat2.9.3.2. (Goudet, 1995).

To test for occurrence and extent of spatial genetic sub-structuring we used an individual level principal component analysis (PCA) using Smartpca as implemented in Eigensoft 4.2. (Patterson, Price & Reich, 2006). Principal components analysis summarizes the variance in microsatellite data into a smaller number of orthogonal components, and the analysis is free of assumptions on the clustering of individuals, nests, localities or populations. Major genetic structuring is captured in the first few principal components, and an implemented Tracey-Wisdom statistic indicates if the components summarizes significant substructure.

Genetic differentiation among subpopulations was further examined by calculating global and pairwise Fst’s (Weir & Cockerham, 1984) for each locality and each colony using Fstat. The R package Hierfstat v. 0.04-10 (Goudet, 2005) was used to estimate hierarchical population substructure at the following levels: individual, nest, locality and total population. Hierfstat achieves this through the computation of variance components (hierarchical F-statistics) in a fully random hierarchical design. IBD patterns were tested with Mantel tests (9,999 permutations), with the package cultevo for R (Stadler, 2017). IBD was tested both among localities within the total population and among nests within the total population.

Analyses on a single individual per nest

Bayesian assignment analyses

We ran a Bayesian clustering analysis on the data by using the software structure 2.3.4 (Pritchard, Stephens & Donnelly, 2000), which clusters individual genotypes by probability of similarity, regardless of the origin of the individuals. In case that nest kin structure violates the underlying model assumptions, we used only a single individual per nest. The most likely number of clusters (K) was estimated by running the analysis with K ranging from 1 to 12, using the admixture model with locprior and correlated allele frequencies. For each K value, the analysis was run ten times with a burn-in of 100,000 steps for a total run length of 300,000 steps. Structure Harvester (Earl & Bridgett, 2012) was used to plot the mean and standard deviation of L(K) for each run, and to apply the deltaK method (Evanno, Regnaut & Goudet, 2005) to determine the most likely number of clusters in the data set.

Mitochondrial population structure

Mitochondrial sequences were trimmed, edited and aligned using CodonCode Aligner (CodonCode Corporation, Dedham, MA, USA). To visualize haplotype divergence, a Neighbour-joining tree (rooted at the centre) using Jukes and Cantor distance and 1,000 bootstraps was constructed in SeaView 4.2.12. (Gouy, Guindon & Gascuel, 2010). Genetic differentiation among sub-populations was estimated by calculating overall FST (FST-mt) using haplotype frequency data in Arlequin v. 3.5.1.2 (Excoffier & Lischer, 2010). A test for IBD among localities was performed on pairwise FST’s calculated in Arlequin as above.

Sex-biased dispersal

Mitochondrial haplotype and microsatellite profiles from one and the same individual per nest were used to test for sex differences in dispersal. Because of haplo-diploidy and smaller effective population size in the mitochondrial compared to nuclear genome, male and female gene flow match when: 14×ΦST=23×FST(1)

assuming Wright’s infinite island model (Berg, Lascoux & Pamilo, 1998). The infinite island model makes several strict assumptions about dispersal, demographic structure and genetic system (Wright, 1931), and its use has been strongly criticized (Whitlock & Mccauley, 1999). However, for ants the use of these estimates appears reliable. In species where queens are wingless (and disperse only by foot) and males have wings, the expected difference in dispersal capability is accurately reflected in the results of the analysis (Seppä & Pamilo, 1995; Fernández-Escudero, Seppä & Pamilo, 2001; Fernández-Escudero, Pamilo & Seppä, 2002; Seppä et al., 2006; Berghoff et al., 2008).

Results

Analyses using the complete data set

Only individuals with microsatellite genotypes from a minimum of four loci were retained in the analysis, in total ca. 95% of genotyped individuals. Locus FY7 was also excluded from analysis due to inconsistent genotyping and the likely presence of null alleles. The remaining loci had on average 6.25 alleles per locus (SD 2.82; range 2–11). Most loci were in HWE in most localities after Bonferroni correction, and only a few significant deviations appeared randomly: FY13 in BAC, FY7 in FOR and FL12 in MOS. Significant LD found appeared also random; locus FE19 was found to deviate from LD in six locations, but always with different loci, and some other locus combinations were sporadically significant in different locations. We hence kept eight loci for further analyses.

The global inbreeding value (FIS) was estimated at −0.003, suggesting random mating at the level of the total population. FIS estimated by locality ranged from −0.172 to 0.196, with small deviations from zero in eight of the 12 populations (Table 1).

Principal component analysis indicated no significant substructure (PC1: eigenvalue 26.78, Tracey-Wisdom 0.18, PC2: eigenvalue 25.83, Tracey-Wisdom 0.07). A plot of the two first principal components clearly shows the absence of a spatial pattern (Fig. 2).

Figure 2 Factor scores from first two principal components from an individual-based principal component analysis on F. fusca genotype data.

Each dot represents an individual colour-coded by locality.

The overall genetic differentiation among localities was significantly greater than zero in nuclear markers (FST = 0.048; 95% CI [0.030–0.074]). When nest structure was taken into account in a hierarchical F analysis, differentiation among localities within the total population was weaker but still significant (FCT = 0.015). A negative inbreeding value for individuals within nests (FIS = −0.38) was most likely a consequence of the non-independence of individuals sampled from the same nest, i.e. nestmates are related. Finally, the large and similar inbreeding values for nests in the total populations (FIT = 0.31) and for nests in localities (FIC = 0.30) are consistent with a strong Wahlund effect, i.e. that nestmate workers are related and that nests act as if they are randomly breeding subunits within the localities. These results hence suggest that nests are highly differentiated within localities and in the total population, and that there is only very weak substructure corresponding to geographic location, results which are in line with those obtained from the principal components analyses.

We found no IBD pattern among localities (r = 0.13, P = 0.214), nor among colonies (r = 0.0183, P = 0.651). However, several pairwise FST’s between localities were significant (Table S2).

Analyses of a single individual per nest

Bayesian assignment analysis

The Structure analysis had the highest log likelihood for K = 2 (Fig. S1), and the DeltaK method also suggested K = 2 (Fig. S2). On examination of the assignment plot (Fig. S3) it was apparent that every individual was assigned to both clusters approximately equally, suggesting no substructure according to geography in the dataset.

Mitochondrial population structure

A total of 555 bp of COII sequence data was recovered for mtDNA analysis. There were 25 informative differences in the dataset (position: 60 (C/T) 69: (C/T), 78: (A/C), 123: (T/C) 144 (C/T), 156 (A/G), 228(C/A), 319 (T/C), 351 (G/A), 354 (C/A) 537(T/C)), which resulted in eleven different haplotypes. The NJ tree resolved two fairly well separated clades, joined by an intermediate haplotype (Fig. 1). The most common haplotype was A (66.8% of the sequenced individuals), followed by haplotypes I (19.3%), D (4.6%) and L (4.0%). The remaining haplotypes (B, C, F–H, J, K) were present in 1% or fewer individuals, but all haplotypes were found in more than one individual. Several haplotypes (C, B, D, F, G, and L) differed only in one base from the A haplotype. More than one haplotype was present in all localities. The locality MAT had the highest number of haplotypes (five), in its 13 sampled colonies, and the average number of haplotypes across all localities was 2.75 (Table 1). The spatial distribution of mitochondrial haplotypes among localities appeared random (Fig. 1) and this pattern was mirrored within localities (plots not shown). Mitochondrial differentiation (FST-mt = 0.018) was not significantly larger than zero, suggesting panmixia. The test for isolation by distance was not significant (r = −0.0415, P = 0.573).

Sex-biased dispersal

The overall genetic differentiation among localities estimated from a single individual per nest was larger for the microsatellite genotype data (FST = 0.040; 95%, CI [0.011–0.076]) than for the mtDNA data (see above), and only the former was significantly greater than zero. After adjustment (Eq. 1), the value indicating male gene flow (adjusted FST = 0.027) is six times larger than that indicating female gene flow (adjusted FST-mt = 0.0047).

Discussion

Our study of the spatial genetic structure of populations of the pioneer ant species F. fusca in southern Finland has produced two key results. First, both nuclear and mitochondrial differentiation among populations are weak and there is no evidence for dispersal limitation. This suggests that the total population is very weakly structured across the entire ca. 35 km2 study area. Second, both female and male gene flow is very high, with a trend for higher female dispersal.

In line with expectations for independently founding ant species, mitochondrial data showed that the localities were colonized by several different and independent matrilines. Correspondingly, there was neither significant differentiation in mtDNA nor any IBD. These results suggest high female dispersal and a panmictic population substructure (Oomen et al., 2011). Our results may even underestimate queen dispersal because the estimates are based on only one sequenced individual per nest. Nest-mate queens are sometimes unrelated (Hannonen, Helanterä & Sundström, 2004; Bargum, Helanterä & Sundström, 2007) and by sampling and sequencing more individuals per nests higher haplotype diversities per locality might have been revealed. Nevertheless, queen dispersal is undoubtedly high in this species.

High dispersal was also evident from the nuclear data. Bayesian clustering and PCA suggested that the total population is homogenous and overall FIS suggested random mating in the total population. Furthermore, the absence of IBD showed that genetic substructure did not arise with increasing distances either; hence dispersal was not limited across the study area. Patterns arising from nuclear DNA reflect male and queen substructure combined, both confirming high dispersal in queens and suggesting high dispersal also in males. In ants, weak population substructure or panmixia over similar spatial scales has earlier been observed in the monogyne form of Solenopsis invicta (Ross & Shoemaker, 1997) and in several species of wasp (Queller, Strassmann & Hughes, 1992; Strassmann, Queller & Solís, 1995; Hoffman, Kovacs & Goodisman, 2008) showing similar or higher rates of dispersal in social Hymenopterans with independent nest founding. Our results for F. fusca probably do not extend to its entire distribution, however, and dispersal barriers such as large water bodies or other dispersal restriction may give rise to genetic differentiation at greater spatial scales.

By testing for sex-biased dispersal, we found that gene flow was six times higher in the mitochondrial than in the nuclear genome, suggesting female-biased dispersal in F. fusca. Of the 15 ant species where sex-biased dispersal has been earlier investigated with the same methods (Table S1), only F. fusca clearly shows higher queen than male gene flow, and only seven comparisons were greater in magnitude than those for F. fusca. Of the latter comparisons, four were on species where queens have no or altered wings and normal winged males. We note that the power of our analysis is not very high given the high dispersal rates in both queens and males.

An earlier study on F. fusca found that colony fathers (genotyped spermathecal content) were related in an older habitat, but unrelated in a younger habitat (Hannonen, Helanterä & Sundström, 2004), without corresponding changes in queen relatedness. If queens mate before they arrive at the habitat patch, they carry sperm from (unrelated) males in their spermathecae, which would make the male population genetically diverse in young habitat. Previous research has shown annual nest survival rates of about 65% (Bargum & Sundström, 2007), ensuring both that genetic diversity from the founding queens is maintained for several years after the initial colonization, and that there is a constant supply of vacant nest sites. Soon after colonization, daughter queens are produced (Bargum, Helanterä & Sundström, 2007) that local males can mate with. Local mating between males and daughter queens in polygyne nests is expected to increase relatedness among colony fathers with time. This is in line with the findings in (Hannonen, Helanterä & Sundström, 2004) and should result in a greater spatial genetic structure among males than among females, as we report here. If vacated nests sites are predominately colonized by immigrant queens, genetic diversity in the queen population may remain relatively stable across time, despite local mating by males. When our present results and those from earlier studies in F. fusca are considered together, female-biased dispersal seems likely in F. fusca. Although the general pattern reported from population genetic studies on sex-biased dispersal suggest that male-biased dispersal dominates in ants (Table S1), these studies are few, and show considerable taxonomic and physiological bias. Female-biased dispersal is assumed for species with wingless males, but its actual prevalence among ants with winged sexuals is unknown.

Formica fusca queens are physiologically capable of long distance mating flights. The fat content of F. fusca is sufficiently high (ca 50% of bodyweight) to ensure good resources for mating flights (Keller & Passera, 1989). The average weight of F. fusca queens is ca 22 mg (range 10.19–30.54 mg; S. Hakala; 2006–2007, unpublished data). Following Helms et al. (2016), ants within this weight-range fly about 35–45 m above ground. F. fusca flight altitude is not known, but the similar-sized and closely related F. lemani (Goropashnaya et al., 2012) flies at an altitude of 40 m (Duelli, Naäf & Baroni-Urbani, 1989). Since F. fusca’s physiology apparently allows for dispersal but flying potentially occurs at low speeds, wind conditions must be favourable for longer distance dispersal (Markin et al., 1971). At an altitude of 50 m, the average wind speed in the sampled area during the mating flight season ranges between 3.80 and 5.60 m/s (http://www.tuuliatlas.fi/en/). Travelling the shortest distance between the localities sampled for this study (1.4 km) would take 4–6 min on such wind currents, and the farthest distance (57.2 km) 170–250 min. This is a simplification of insect flight and wind dynamics, but in short, dispersing F. fusca sexuals could potentially reach considerable dispersal distances in our study area given a flying altitude of ca 40 m and normal wind conditions.

The PCA analysis and the hierarchical F-statistics revealed nuclear genetic substructure only at the level of the nest. This pronounced colony-level structure is consistent with monodomy and significant within-colony relatedness, as reported from previous studies (Pamilo, 1983; Hannonen, Helanterä & Sundström, 2004; Bargum, Helanterä & Sundström, 2007). Finally, nuclear differentiation among sampling localities, as indicated by the low FST estimates, was weak, but significantly greater than zero. Thus, the study area was not genetically entirely homogenous, but there was no pattern of IBD, neither among localities, nor among nests across localities, in neither of the markers. Such a pattern is expected to arise when dispersal is high in a spatially and temporally unstable habitat (Wade & McCauley, 1988; Leblois et al., 2000; Yang, Bishop & Webster, 2008; Peel et al., 2013), such that the number of long-distance colonizers founding new populations is at least twice the number of local colonizers. The scenario is consistent with the ecology of F. fusca, since optimal habitats are patchily distributed and rapidly colonized (Punttila et al., 1991). These habitats are also temporally variable. Continuous colonization by F. fusca into clear-cut plots leads to a steady increase in nest densities for several years (Punttila et al., 1991). As the forest succession advances dispersal remains high but the habitat becomes less suitable for F. fusca, and more suitable for mound-building wood ants (Formica sensu stricto) who arrive and outcompete them (Punttila et al., 1991). Similar population genetic substructure has been reported for other invasive and pioneer species that colonize plots in a similar fashion to F. fusca such as Vaccinium plants (Yang, Bishop & Webster, 2008), the invasive cane toad (Leblois et al., 2000) and the invasive yellow crazy ant Anoplolepis gracilipes (Drescher, Blüthgen & Feldhaar, 2007).

Conclusion

This study shows high male and queen dispersal, and as a consequence, near panmictic population genetic structure in F. fusca. Our results are consistent with independent founding and the winged status of both sexes, and with species’ ecology. Temporally and spatially variable habitat necessitates the maintenance of wings and independent founding for rapid colonization and habitat escape, in both queens and males. Polygyny in F. fusca may nevertheless be a dispersal adaptation. Re-adopted queens face no dispersal risks, and if unrelated queens can also either be adopted into nests, or if queens can found nest through pleometrosis, dispersal risks are reduced also for dispersing queens. The results in this study contrast with findings from other polygyne European Formica species, for which stable habitat and weak dispersal are thought to promote polygyny and polydomy (Rosengren & Pamilo, 1983). They also contrast with results for highly invasive ant species, for which human-mediated dispersal and highly unstable habitats are closely associated with polygyny and supercoloniality (Holway et al., 2002). Relative to these, F. fusca’s habitat is intermediate in stability, and perfectly exploited by queen dispersal strategies.

Supplemental Information

Supplemental Information 1 Sex biased dispersal in ants, pairwise FSTs and bioclimatic variables.

Click here for additional data file.

Supplemental Information 2 11 mtDNA COI haplotypes for F. fusca (fasta file).

Click here for additional data file.

Supplemental Information 3 Microsatellite data for F. fusca.

Click here for additional data file.

Supplemental Information 4 L(K), deltaK and assignment probabilities from STRUCTURE clustering.

Click here for additional data file.

We are grateful to Katja Bargum, Stephen J. Martin, Riitta Ovaska and Anniina Mattila for help in collecting samples, and Soile Kupiainen for genotyping. Sanja Hakala kindly shared her weight measurements for F. fusca queens.

Additional Information and Declarations

Competing Interests

Author Contributions

Data Availability

The authors declare that they have no competing interests.

Helena Johansson performed the experiments, analysed the data, prepared figures and/or tables, authored or reviewed drafts of the paper, approved the final draft.

Perttu Seppä conceived and designed the experiments, performed the experiments, authored or reviewed drafts of the paper, approved the final draft.

Heikki Helanterä conceived and designed the experiments, performed the experiments, contributed reagents/materials/analysis tools, authored or reviewed drafts of the paper, approved the final draft, collected samples.

Kalevi Trontti conceived and designed the experiments, performed the experiments, authored or reviewed drafts of the paper, approved the final draft.

Liselotte Sundström conceived and designed the experiments, performed the experiments, contributed reagents/materials/analysis tools, authored or reviewed drafts of the paper, approved the final draft.

The following information was supplied regarding data availability:

The raw data are provided in the Supplemental Files.

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
