# Peer review of "Weak population structure in the ant Formica fusca"

_PeerJ, doi:10.7717/peerj.5024_

## Round 0.1 · original submission · Major Revisions

Two reviewers have assessed this manuscript. Both agree that the subject is interesting and that the paper may be of interest to readers. However, there are a number of issues that would need to be addressed. Reviewer 2 has provided particularly detailed comments and these should all be addressed.

I also had trouble following the logic in the introduction and I think this needs to be rethought and rewritten to make arguments more clearly.
Also please be clear about when you used multiple individuals from a colony and when you used only a single one. In fact, is there are reason to use more than one for any analysis, given the pseudoreplication issue, also pointed out by reviewer 2.

When looking for IBD, please use a Mantel test or autocorrelation analysis, as a linear regression is not valid as the point are not independent.

Please also consider carefully the other suggestions of reviewers.

Reviewer 1 ·

Basic reporting

no comment

Experimental design

no comment

Validity of the findings

no comment

Additional comments

The authors report an interesting study on an ant inhabiting unstable/recently available habitats in Finland. It expands previous knowledge on the population genetics of Formica fusca at a higher geographical scale using numerous and adequate analyses. The main result of the manuscript comes from the fact that for the first time a female-biased dispersal rate is reported for an ant species. The reading is fluent and the language is clear. I would recommend its publication in PeerJ after the authors correct some reference issues and consider some of my comments below:

- In lines 317-331 the authors comment about young and old habitats, suggesting a male-biased dispersal in the young and a female-biased in the old. According to their results of panmixia in the locations analysed, these would be then considered as old but there is no formal statement about this in the text. In a previous work from some of the authors (Hannonen, Helanterä & Sundström, 2004), they clearly reported the age of the populations studied but in the current study it is not mentioned. Do the authors know the age of the sampling localities or just guess that they should be old due to results of female-biased dispersal?

- In the discussion the authors also state that the highest dispersal is expected in new optimal habitats whereas subsequent colonization would lead to higher nest densities. Then, in the latter situation long distance dispersers would not be likely to occur, which could be interpreted as a change in male strategy: reduced dispersal as a way to increase their mating opportunities. Reading the manuscript one could get the idea that the females would actively disperse at higher rates than males in old habitats but maybe it should be worth discussing a bit more about this under the male’s perspective.

- The format of the references should be seriously checked as I found numerous mistakes: PeerJ journal states that in-text citations of four or more authors should be abbreviated with the expression “et al”; however the expression “& al” can be observed throughout the text. Also, the comma before the year is also missing in several references. In the case of three authors, these should be named completely but, however in the manuscript I could find a few times the same reference cited in two ways, one according to the journal rules and another as first author et al. (e.g. Bargum et al., 2007; Sündstrom et al., 2005). In other cases, three author references are only treated as first author et al. (e.g. Sanllorente et al., 2015; Gyllenstrand et al., 2004; Drescher et al., 2007).

- Figure 1: The word “text” appears in the middle of the map, it should be removed. Also in this figure, the haplopyte H (yellow color in the NJ tree) is not represented in any nest or at least I am unable to distinguish it.

Reviewer 2 ·

Basic reporting

The paper investigates the population genetic structure of the pioneer facultatively polygynous ant Formica fusca at a large spatial scale (35 km2) using 8 microsatellite markers and a mtDNA sequence. In ants, breeding systems (monogyny or polygyny) is often associated with the mode of colony foundation (dependent or independent colony foundation) and thereby with female dispersal abilities. At small spatial scales (hundreds of meters of few kilometers), restricted female dispersal is known to lead to population viscosity with dispersal mainly biased towards males. The expected pattern of population genetic structure is less clear for independent founding species. In this study, the authors found a merely absence of population substructure for both types of markers even though the population genetic differentiation was slightly significant for microsatellite markers.
I found the manuscript clearly written and the analysis seems appropriated (though see some in the section experimental design). I however identified three main weaknesses (1) in the way the research questions are defined (see my commentin the experimental design), (2) in the choice of the sampling design to assess population genetic structure at large scale (see comments in the experimental design) and (3) in the interpretation of the results that conclude to panmixia in the species (the main conclusion given in the title) (see comments on the validity of the findings).

Experimental design

The hypotheses being tested are not clearly developed in the last paragraph of the introduction and the rationale of the study is not that straightforward to me as explained in the following comments:

2.a. The introduction focuses only on the biological traits of ant species that could affect population structure with the idea that polygyny is more often associated with dependent colony foundation and therefore with restricted female dispersal. This was indeed shown by Keller (1991 in Ethology, Ecology and Eovlution) but using only 24 European ant species and with some exception. This association allows the authors to claim at the end of the abstract that the nearly absence of population genetic structure was unexpected in this facultatively polygynous ants. However, knowing that females are winged and found new colonies independently in this species, I do not see why the results are unexpected even if the species is slightly polygynous. More importantly, at large spatial scales, landscape connectivity, geographical barriers, population demography and colonization history can be more important in shaping the population genetic structure than species dispersal abilities. Moreover at large scale, the influence of genetic drift is strong relative to gene flow (Hutchison & Templeton 1999). Hence, if the aim of the study was to assess dispersal abilities of the species (which is the main focus of the introduction), I do not think that the large spatial scale is the more relevant scale to consider. May be it would be better to introduce earlier the idea that even though both males and female are winged, ants are not very good flyers. Dispersal of sexuals might then be sufficient to maintain panmixia at small scale but not necessarily at large scale.

2.b. If a higher genetic structure could indeed be expected at larger than small spatial scale (as I noted above), it is however not clear to me why this should be due to social organization as mentioned line 141. The authors should make the expected pattern more precise by indicating for instance whether they know the level of polygyny in the populations sampled. Given that in a population with both monogynous and polygynous colonies, no structuration was found at a small scale, why would the effect of social organisation be more pronounced at a larger scale?

2.c. I have the same concern with sex-biased dispersal. Given that sex-biased dispersal was not observed at a small spatial scale (which is in agreement with the fact that both sexes are winged), why would the authors expect sex-biased dispersal at a larger scale? Are there some biological reasons to believe that the scale of sampling will change the results? Indeed at a large spatial scale, as noted above, extrinsic factors will be more important than dispersal abilities in shaping the population genetic structure. For now, it seems that the authors just emphasized this point because of these results (they found a slightly higher dispersal in females, a pattern rarely found in ants). I doubt that they would have addressed the question of a change in sex-biased dispersal at larger scale if they had not found this slightly higher female dispersal.
I would therefore advise the authors to clearly introduce the different forces that could shape the population genetic structure at large spatial scale without focusing only on the species dispersal abilities and to provide a more rational and precise expectation of which pattern of population genetic structure would be expected in the species studied.

2.d. A total of 443 individuals was sampled (Table 1) out of 93 colonies from 12 populations with most populations having less than 10 colonies sampled. In some way, this sampling does not allow to properly assess social organization since five workers is not enough to get precise estimate of within colony relatedness. Given that the aim was to conduct a population genetic study, it would have been more relevant, for the same sample size, to analyze more colonies with a single worker per colony. Because a previous study shows that colony relatedness could be as high as 0.8, workers from the same colony cannot be considered as independent. The low number of colonies sampled by locality probably decreases the validity of the conclusions (see below in section 3).

Validity of the findings

The main conclusion of the paper, emphasized by the authors in the title (line 42-43), the abstract (line 42-43) and the discussion (line 289-291, line 296, line 303, 304), is that at the geographical scale analyzed (35 km2), dispersal is not limited and the population is panmictic across the entire study area. I do not doubt that dispersal should be quite important (as expected since both sexes are winged) but I questioned the fact that mating is panmictic at this scale. This is bit overstated for various reasons:

3.a. First, the effective sample size is quite low given that many workers can be considered as sort of pseudoreplicate as they belong to the same colony and thereby likely share part of their genes. Using only 8 microsatellite markers and such a small number of colonies within localities might not represent a sufficient sample size to properly test for deviation from panmixia.

3.b. Second, it is somewhat surprising to claim that there is panmixia whereas a low but significant genetic structuration was detected with the microsatellite markers (but given my above comment, it is difficult to be confident with these small sample sizes).

3.c. Last but not least, as already mentioned above, it is important to consider other factors than dispersal to explain the results. For instance, because the species leave in temporally instable habitats (line 124), recent colonization could also explain the lack of structuration and the absence of isolation by distance pattern (line 307-308) as populations might not be at migration/drift equilibrium. Basically, the pattern of population genetic structure (whatever the genetic marker considered) does not only reflect present gene flow but also the demographic history of populations.
3.d. The discussion gives too much importance to this idea of panmictic population (line 289-291, line 296, line 303, 304, line 307-315, line 353).

I have few other comments on the analysis:

3.e. line 188 why using a linear regression model (lm) for testing isolation by distance? This might be a problem because pairs of localities (or pairs of colonies) are not statistically independent, a Mantel test (specifically testing association between two matrices) or Moran’s I spatial correlograms are usually preferred (see for instance Diniz-Filho et al 2013, Genet. Mol. Biol.), even though I reckon that these technics can also be discussed (see for instance see Guillot & Rousset 2013 Meth Ecol Evol). Anyway I do not think that this would change the results but the authors should argue their choice of using a classic linear regression model for testing the correlation between two matrices whereas it is not the most appropriate test.

3.f. Information on the landscape surrounding the localities sampled could be useful to better understand whether it is homogeneous and would support the fact that ants could indeed fly across it or whether there is some kind of heterogeneity. I do not ask for a landscape genetic approach as I do not think the sample size is large enough to conduct such analysis but it would be good for the reader to have information about it. May be it is just completely homogeneous and favorable to ants which would confirm that dispersal is not limited by the landscape in this area and be agreement with the results.

3.g. The software STRUCTURE might not be very good at finding population structure with few microsatellite loci and few individuals (see original paper by Pritchard et al 2000). In such situation, subtle population structure can sometimes be more easily detectable when geographic information is used. Given that nests were geolocalised, why the authors did not try to consider geographic information either directly in STRUCTURE or in other software just as GENELAND?

3.h. The authors reckon that the formula used to estimate sex-biased dispersal (line 215), based on the Wright’s infinite island model has been criticized. However, they still use it and mentioned in the discussion (line 317) that they “directly test for sex-biased dispersal” even though they again reckon in the following sentences that violations of the assumptions of the island model might biased the results in an unpredictable way. So I think it is quite dangerous and not very logical to use a formula known to be potentially wrong to draw conclusion on a female-biased gene flow.

Additional comments

In addition to the main comments made in the above sections, I have some minor comments:

4. Line 67 to 70: the recent review by Cronin et al. 2013 in Annual review of Entomology should be cited here.
5.Line 81- 82: this sentence needs to have references to support the statement that substructure at the level of a population is often observed.
6. Line 96-97: please reconsider the structure of the sentence. As such the meaning is not that clear. Do the authors mean that when both mode of colony foundation exist in the same species, few winged dispersal queens may be enough to allow efficient gene flow?
7. Table 1: If the level of Fis given in the table estimated on workers without considering the colony structure?
8. The authors should make clear in the M&M that only a single individual per nest was used for estimating genetic differentiation among localities (as written in the result section line 279). How was the individual sampled (at random, only once ?)? Why showing both the Fst results on all individuals (line 243-244) and on a single individual per nest? Given that individuals from a given nest cannot be considered as independent, why not only presenting Fst based on the reduced sample? This was quite confusing to me. Moreover, when using STRUCTURE, having too many related individuals could also pose a problem.
9. Concerning STRUCTURE: Did the independent runs for each K give consistent results (which would support the idea of an absence of structure)? Finally, Pritchard et al (2000) reckon that it can be difficult to obtain reliable estimates of Pr(X|K) so that just examining them might not be the best way of determining the most likely number of clusters in the data (as done by the authors line 197). Evanno et al (2005 Mole Ecol) proposed a method for choosing the best number of K. Why not having use it? I reckon that given the results (no population structure), deltaK would not be informative. Anyway, to be convincing the authors should provide at least a graph showing the mean and variance of the log likelihood of the different simulations for each value of K tested.
10. Line 331: it is not clear to me why female-biased dispersal may be common among ants? And I do not think that female-biased dispersal is assumed often for monogynous species? On which basis are done these assumptions? Do the authors have references to add here in order to support their statement?
11. In the supplementary file containing the genotypes, there are information in column AA, AB and AC and after the last line of genotypes. Is it a mistake or does it mean something?
12. In the table S1: there is a mistake for C. cursor, independent should be replaced by dependent.

---

## Round 0.2 · accepted · Accept

The authors have done an excellent job of revising their paper in the light of reviewer comments.

#